# Geometric Attention Networks for Small Point Clouds

## Abstract

Much of the success of deep learning is drawn from building architectures that properly respect underlying symmetry and structure in the data on which they operate—a set of considerations that have been united under the banner of geometric deep learning. Often problems in the physical sciences deal with relatively small sets of points in two- or three-dimensional space wherein translation, rotation, and permutation equivariance are important or even vital for models to be useful in practice. In this work, we present an architecture for deep learning on these small point clouds with rotation and permutation equivariance, composed of a set of products of terms from the geometric algebra and reductions over those products using an attention mechanism. The geometric algebra provides valuable mathematical structure by which to combine vector, scalar, and other types of geometric inputs in a systematic way to account for rotation invariance or covariance, while attention yields a powerful way to impose permutation equivariance. We demonstrate the usefulness of these architectures by training models to solve sample problems relevant to physics, chemistry, and biology.

## Introduction

Deep learning has been immensely successful in solving a wide range of problems over the last several years, driven in large part by identifying appropriate ways to embed structure of data and symmetry of problems directly into the architecture of the network—an idea at the core of geometric deep learning[1]. Some applications of geometric deep learning include the use of convolutional filters in CNNs to attain translational equivariance, or graph convolutions in graph neural networks for permutation equivariance.[1] Building symmetry into the architecture of a deep neural network can improve the data efficiency of the network and guarantee important analytical properties without having to rely on the network to learn to approximate them from training data.

In this work, we derive a family of architectures that is useful in applications from physics to biology, where problems often deal with relatively small point clouds of labeled coordinates. These could be local environments of particles assembling into a crystal[2], atoms in a molecule interacting with other atoms[3], or coarse-grained beads representing parts of a protein[4]. In many of these applications without the influence of an external field, we are interested in modeling attributes of the

---

[1] In this work, we use the following terms to discuss symmetry of functions $f$ and operations $\rho$: $f$ is *invariant* to $\rho$ if it does not change when $\rho$ changes: $f \circ \rho = f$. If $f$ and $\rho$ commute, then we say that $f$ is *covariant* with respect to the operation of $\rho$: $f \circ \rho = \rho \circ f$ (some sources call this equivariance or same-equivariance; the typical definition of equivariance is more general, but we will only discuss $f$ and $\rho$ that are endomorphisms in the context of covariance). Here we use *equivariance* to broadly mean considerations of covariance as well as invariance (since scalars of interest are typically invariant to translation and rotation in physical applications) for simplicity of discussion.

Submitted to 35th Conference on Neural Information Processing Systems (NeurIPS 2021). Do not distribute.

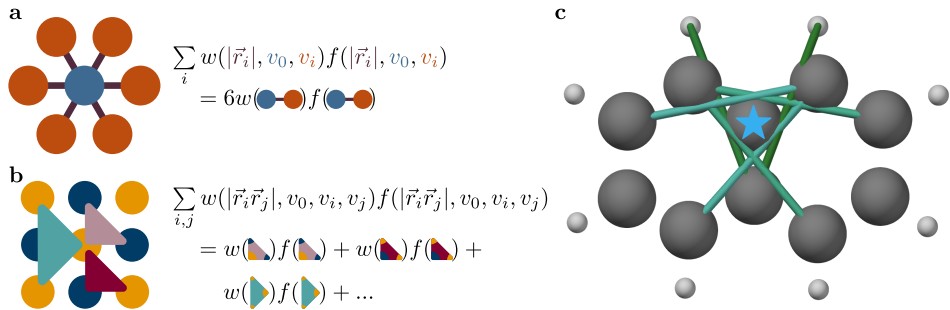

Figure 1: Overall strategy for incorporating rotation and permutation equivariance into deep neural networks using attention mechanisms and geometric products. (a) At its simplest level, our proposed structure uses an attention mechanism over the bond lengths of a cloud of points, each of which carries a value as commonly used in graph neural networks. (b) Geometric products (or linear combinations thereof) can be used to combine pairs, triplets, or larger tuples of vectors in a systematic and geometrically meaningful way. Rotational equivariance can be attained by using invariant or covariant quantities, as desired. (c) An attention mechanism reduces the set of generated geometric products to enforce proper permutation equivariance, and the learned attention maps can provide insights into how models operate. For example, here, a carbon atom in a naphthalene molecule (indicated by a blue star) directs its focus broadly around the carbon atoms of the aromatic rings in which it is situated, rather than focusing exclusively on its nearest neighbors in the molecular graph. Brighter-colored bonds indicate a greater attention weight for the two atoms sharing the bond.

system—such as the identity of a particle's local self-assembly environment, or the potential energy of a group of atoms—which are invariant with respect to rotation of the input coordinates, as well as permutation in the ordering of points. Here we attain rotation invariance by constructing functions from rotation-invariant components of geometric products of input vectors from geometric algebra, and permutation invariance by using an attention mechanism to intelligently reduce representations over the set of vector products.

## Related Work

**"Large" point clouds.** Point clouds are a ubiquitous data structure and are often found in domains outside of the physical sciences. For the purposes of this work, we focus on comparatively small sets of points where the points are relatively information-rich—for example, carrying information about atom identities, local environments, or other information—in contrast to point clouds commonly found in computer vision and robotics which may represent the geometry of a mesh or be sampled from an object, but otherwise not have as much information associated with each point. We refer to Guo *et al.* for a survey of this field[5], but a few of the recently-developed notable approaches include PointNet[6], deep sets[7], and kernel point convolutions[8].

**Geometric approaches for small point clouds.** Many architectures have been proposed to incorporate rotation equivariance by augmenting graph neural networks with geometric attributes that are known to be rotation-invariant, such as bond lengths and angles. SchNet[9] learns distance-based convolution filters which are used to propagate signals over graphs. PhysNet[10] also refines node representations based on bond lengths, while incorporating a learnable attention mechanism. DimeNet[11] extends the information used to calculate node-level representations to include angles between bonds. GNNFF[12] generates rotation-covariant results by computing a weighted sum of modulated input vectors based on a graph message passing scheme.

**Group representation-based approaches.** These methods take advantage of group representation theory by first transforming inputs into a space in which rotation-equivariant maps are more easily expressed. This set of methods is powerful, having been used in the past to design rotation- and permutation-equivariant models[13–15], and have even been expanded recently for arbitrary groups[16]. Attention-based models have also been utilized in this area: SE(3) Transformers[17]

extend tensor field networks[14] with a self-attention mechanism for increased expressivity by incorporating value- and geometry-dependent attention weights.

The approach we present here is similar to several of the ideas presented above; however, rather than specifying particular rotation-invariant quantities to utilize or learning maps that operate on irreducible representations, we leverage the structure provided by geometric algebra to determine which rotation-invariant and -covariant quantities are of interest.

## Geometric Attention Networks

In this work we formulate deep neural networks using learnable functions consisting of two parts: (1) a set of geometric products of input vectors; and (2) a permutation-equivariant reduction over these products using an attention mechanism. We describe each of these aspects below.

### Geometric Algebra

The geometric algebra was developed in the 19th century and provides a consistent framework for dealing with scalars and other geometric quantities—such as vectors, areas, and volumes in three-dimensional space—in arbitrary dimensions[18]. Here, we will describe the essential parts of geometric algebra as related to our proposed attention mechanism, and defer to other works for a more thorough description[19]. The geometric algebra specifies a binary operator, the *geometric product*, that works on *multivectors*. Multivectors can be expressed as linear combinations of terms from a fixed basis set for a given space, such as $\mathbb{R}^2$ or $\mathbb{R}^3$; in three-dimensional space, this yields scalars, vectors, *bivectors* (which specify signed areas within a plane and have 3 components), and *trivectors* (which specify signed volumes and have 1 component)—a total of 8 linearly independent terms for each multivector[2]. When rotation invariance is desired, we can utilize the rotation-invariant components of a multivector: scalars, trivectors, the norms of vectors, and the norms of bivectors are rotation-invariant. As an example, the geometric product of two vectors yields a scalar plus a bivector; the scalar component is the dot product, and the bivector component is related to the cross product of the two.

Geometric algebra provides a general framework that can be used to build up expressive functions as linear combinations and geometric products of multivector inputs; rotation-equivariant quantities can then be derived from the products, depending on the application and symmetry of the problem of interest. The types of elements produced by a geometric product of two multivectors in $\mathbb{R}^3$ with the given components are listed in Table 1 below.

Table 1: Terms arising from the geometric product $AB = (A_s + A_v + A_b + A_t)(B_s + B_v + B_b + B_t)$ in $\mathbb{R}^3$. In three dimensions, multivectors $A$ and $B$ consist of scalars (*s*), vectors (*v*), bivectors (*b*), and trivectors (*t*).

|  | $\mathbf{B_s}$ | $\mathbf{B_v}$ | $\mathbf{B_b}$ | $\mathbf{B_t}$ |
|---|---|---|---|---|
| $\mathbf{A_s}$ | s | v | b | t |
| $\mathbf{A_v}$ | v | s + b | v + t | b |
| $\mathbf{A_b}$ | b | v + t | s + b | v |
| $\mathbf{A_t}$ | t | b | v | s |

From Table 1, we can see that successive products of vectors alternate between producing two types of multivectors: products of even numbers of vectors yield a scalar and bivector ($(v+t)v = vv+tv = (s + b) + b \rightarrow s + b$), while products of odd numbers of vectors produce a vector and trivector ($(s + b)v = sv + bv = v + (v + t) \rightarrow v + t$). Generating rotation-invariant quantities from these products is the primary application of geometric algebra in this work, although in general the method

---

[2]Multivectors form a vector space: the individual components of multivectors (any number of bivectors, for example) can be directly summed elementwise, but multivector components of different types stay separate and are multiplied using the distributive property of geometric products when needed. The geometric product has an identity of the scalar 1 and is associative; in other words, it forms a monoid over multivectors.

could be used to incorporate different types of scalar, vector, bivector, and trivector quantities; for example, rotations could be input as quaternions, which are isomorphic to the scalar-and-bivector product of even numbers of vectors.

**Attention from Geometric Products**

For input point clouds with $N$ points, we can construct a series of successively higher-order geometric products for all $N^2$ possible pairs, $N^3$ triplets, and so on; these individual points, pairs, or triplets we will call a *tuple* in this context. In addition to a coordinate $\vec{r}_i$, we associate a set of values $v_i$ to each point indexed by $i$ in some space with a given *working dimension* (we avoid calling these vectors to decrease the confusion with geometric vectors; these correspond to the non-geometric attributes of the point, such as type embeddings). To create permutation-covariant functions (producing a value for each input point) or permutation-invariant functions (producing a single output value), we make use of a simple attention mechanism based on the rotation-invariant attributes of each tuple. Attention has been used widely in applying deep learning to a range of problem domains over the last few years, with particular success in the field of natural language processing[20]. Since we are already generating tuple-wise quantities, we choose to utilize a simpler mechanism than the typical dot product self-attention. We specify four functions: a value-generating function $\mathcal{V}$, a tuple value-merging function $\mathcal{M}$, a joining function that summarizes the invariant and tuple representations $\mathcal{J}$, and a score-generating function $\mathcal{S}$. The functions have the following uses within the network:

- $\mathcal{V}$ produces features in the working dimension of the model from the invariants associated with each tuple.

- $\mathcal{M}$ merges the 1, 2, 3, or more values associated with a tuple of input points into the working dimension of the model. The form of $\mathcal{M}$ could be a complex function, a learned linear projection for each tuple position, or simply taking the sum of the tuple values.

- $\mathcal{J}$ joins the invariant representations from $\mathcal{V}$ and the tuple representations from $\mathcal{M}$. Like $\mathcal{M}$, it could be a learned projection or a simple sum function.

- $\mathcal{S}$ generates score logits from the representation of each tuple, which incorporates invariants associated with the tuple and the values being associated with each point that is part of the tuple. The results from $\mathcal{S}$, passed through a softmax function, will yield the weights for the attention mechanism.

We first calculate the multivector geometric products $p_{ijk...}$ of all combinations of input vectors $i$, $j$, $k$, and so on, up to a specified rank (pairwise attention would produce a two-dimensional matrix of products $p_{ij}$). We then use $\mathcal{V}$, $\mathcal{M}$, $\mathcal{J}$, and $\mathcal{S}$—together with a function extracting the rotation-invariant attributes of a geometric product (the scalar component, trivector component, and the norms of the vector and bivector components)—as follows for a network producing permutation-covariant outputs $y_i$:

$$
\begin{aligned}
p_{ijk...} &= \vec{r}_i \vec{r}_j \vec{r}_k ... \\
q_{ijk...} &= \text{invariants}(p_{ijk...}) \\
v_{ijk...} &= \mathcal{J}(\mathcal{V}(q_{ijk...}), \mathcal{M}(v_i, v_j, v_k, ...)) \\
w_{ijk...} &= \underset{jk...}{\text{softmax}}(\mathcal{S}(v_{ijk...})) \\
y_i &= \sum_{jk...} w_{ijk...} v_{ijk...}
\end{aligned}
\tag{1}
$$

If a permutation-invariant reduction is desired, then the softmax and final sum can be performed over all tuples simultaneously, rather than for each input point individually. While $\mathcal{J}$, $\mathcal{V}$, and $\mathcal{M}$ could in principle be used to change the working dimension as permutation-covariant layers are stacked on

top of each other, in this work we keep the working dimension constant for the sake of easily adding residual connections.

If rotation-covariant, rather than rotation-invariant, behavior is needed for the output of the network, the same attention structure can be used with slight modifications; here, we coerce a vector from the product $p_{ijk...}$ (which consists of directly taking the vector component from products of odd numbers of input vectors, or multiplying a bivector by the unit trivector to produce a vector—as shown in the last column of Table 1—in the case of even numbers of input vectors). These vectors can be combined with a scalar rescaling each vector—generated by a learned function $\mathcal{R}$—and the attention mechanism to yield

$$\vec{r}_i' = \sum_{jk...} w_{ijk...} \mathcal{R}(q_{ijk...}) \text{vector}(p_{ijk...}). \tag{2}$$

## Results

We demonstrate the utility of our geometric algebra attention scheme by training deep networks to solve three problems appearing in physics, chemistry, and biology. For simplicity, all the models presented here utilize pairwise attention with a working depth of 32 units. Value functions $\mathcal{V}$, score functions $\mathcal{S}$, and rescaling functions $\mathcal{R}$ are simple multilayer perceptrons with a hidden width of 64 units, with layer normalization applied to the output of $\mathcal{V}$. The network for crystal structure identification uses the mean function for merge functions $\mathcal{M}$ and join functions $\mathcal{J}$, while the other two applications use learned linear projections. Networks are trained for up to 800 epochs using the adam optimizer[21]; the learning rate is decreased by a factor of 0.75 after the validation set loss does not decrease for 20 epochs, and training is ended early if the validation set loss does not decrease for 50 epochs. Numerical results are reported as the mean and standard error of the mean over 5 samples. Python code under the MIT license implementing each experimental workflow is included in the supplementary information.

### Crystal Structure Identification

On length scales ranging from those of atoms to colloidal particles, matter often organizes itself into ordered two- or three-dimensional structures. One of the core ideas of materials science is that structure is one of the major determining factors for material behavior. With this perspective in mind, when studying computational models of self-assembling systems we often first identify what structures, if any, have formed in our simulations—a task complicated by naturally-occurring thermal noise, crystallographic defects, and potentially the complexity of the structures themselves. Early efforts to automatically characterize structure led to the widely-used Steinhardt order parameters[22–24], which are rotationally-invariant sums of spherical harmonic magnitudes over local particle environments. While the Steinhardt order parameters can be useful when studying phase transitions or distinguishing among a small number of phases, determining appropriate hyperparameters—including neighborhood size to consider, spherical harmonic order $\ell$ to use, and thresholds to identify behaviors of interest—can be difficult[23]. For this reason, data-driven approaches to analyzing structure have been the subject of great interest in recent years[25].

We use geometric attention networks to identify the source structure type of small neighborhoods of particles extracted from bulk crystals. We select 8 prototypes of single- and two-component crystals from the AFLOW Encyclopedia of Crystallographic Prototypes[26, 27]. These structures are chosen to demonstrate that models can learn not only geometric information ($cF4$-Cu and $hP2$-Mg are similar structures but with a different stacking of their close-packed layers; the clathrates $cP46$-Si and $cF136$-Si are also similar, with a different arrangement of many common motifs), but also the information encoded within each point ($cP2$-CsCl and $cF8$-ZnS differ from $cI2$-W and $cF8$-C only by their particle type assignments). For each structure, we rescale the unit cell such that the shortest nearest-neighbor distance over the structure is 1 before replicating the unit cell to consist of at least 2048 particles. We then create three samples of each structure by adding Gaussian noise

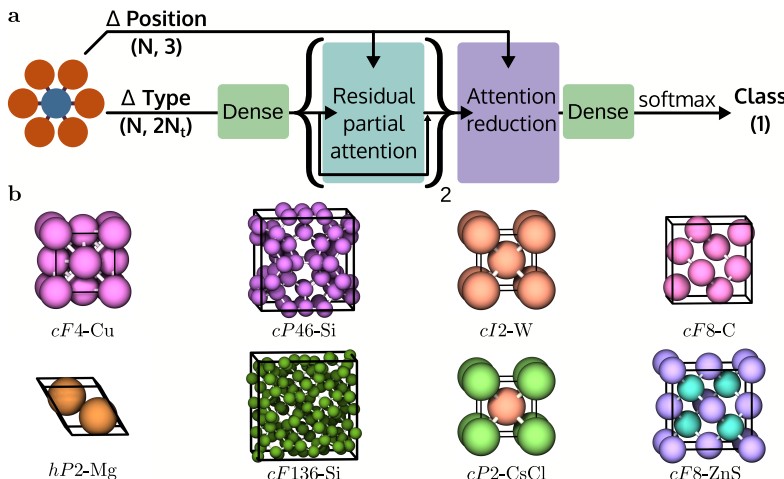

Figure 2: (a) Network architecture for crystal structure identification. Coordinates and particle types are passed through two permutation-covariant layers before a final permutation-invariant reduction. (b) Crystal structure prototypes chosen for the structure identification benchmark. Simple and complex structure types—including two binary structures—are included.

with a standard deviation of $10^{-3}$, $5 \cdot 10^{-2}$, and $0.1$ separately to the particle coordinates, in order to emulate thermal noise. For each particle in the structure, we find the 12 nearest neighbors and their associated types using the freud[28] python library, which we feed into the network as the pairwise distance $\vec{r}_{ij} = \vec{r}_j - \vec{r}_i$ and one-hot-encoded symmetrized type vector $\Delta t_{ij} = [I_{t_i} - I_{t_j}, I_{t_i} + I_{t_j}]$, where $I$ is the identity matrix of dimension corresponding to the maximum number of types.

We train classifiers with 2 permutation-covariant attention blocks before a final reduction over the entire particle neighborhood—as shown in Figure 2—in order to categorize local particle environments according to their source crystal structure type. These networks rapidly learn to identify structures after a few epochs, with a final overall accuracy of $98.7\% \pm 0.2\%$ after training for roughly 45 minutes on an NVIDIA Titan Xp GPU.

**Molecule Force Regression**

One of the most dramatic contributions of deep learning to the field of chemistry lies in constructing fast, accurate approximations of expensive physical calculations[29, 30]. Machine learning models can be many orders of magnitude faster than the methods used to generate their training data, which can bring vastly more detailed and longer-time simulations into the realm of possibility. Central to the applicability of these methods are issues of symmetry and equivariance: any imperfection in rotational invariance of a learned potential energy function could ruin the proper thermodynamic behavior of a model, for example, so models must be carefully designed to ensure physical behavior.

In a method similar to Batzner *et al.*[31], we train models to predict the per-atom forces calculated using *ab initio* molecular dynamics and density functional theory available in the MD17 dataset[3]. As shown in Figure 3, we first transform the raw coordinates and types of each atom in a given molecule into the pairwise difference and symmetric sum and difference of the coordinates and one-hot type encoding for each atom with respect to each other atom, respectively, to fix translation invariance and assign type representations to the pairwise particle bonds. We then perform a series of geometric attention calculations, calculating new values per atom, which are finally summed to produce a scalar energy. The gradient of this energy with respect to the input coordinates is used to produce the force output of the network, which ensures that a conservative force field is learned.

Consistent with previous benchmarks on this dataset, we train networks using the mean squared distance loss for each molecule using 1,000 snapshots of forces each as training, validation, and test

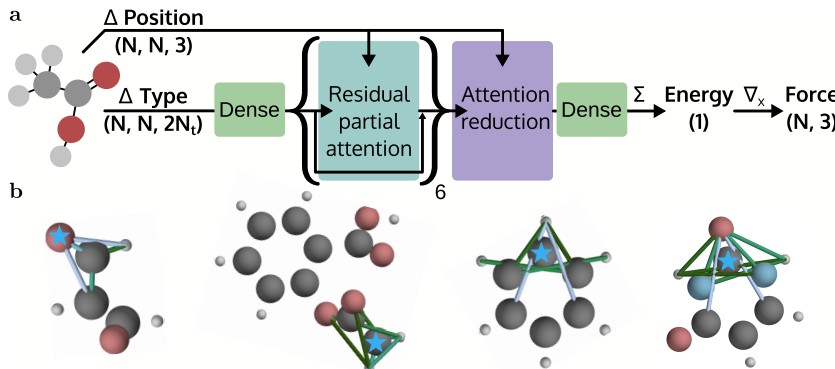

Figure 3: (a) Network architecture for molecular force regression. Coordinates and particle types for all atoms in a molecule are fed into the network as a set of pairwise distances, with the atomic representations refined through a series of geometric algebra attention layers. Six permutation-covariant layers are stacked before reducing the representations with a final, permutation-invariant geometric product attention layer. (b) Sample pairwise attention maps for four training data molecules (malonaldehyde, aspirin, benzene, and uracil) after filtering out low-attention pairs. The attention maps indicate how strongly the pair of atoms joined by the line affect the representation of the atom indicated with a star, with lighter lines indicating greater influence. Qualitatively, more complex bonding environments such as those on the right tend to have longer-range attention interactions than the simpler environments on the left.

data sets. We also report results for models trained on all molecules' data simultaneously, or 8,000 snapshots each for training, validation, and testing. Training a model on an individual molecule's data takes between 30 minutes (ethanol and malonaldehyde, with nine atoms each) to two hours (aspirin, with twenty-one atoms) on an NVIDIA Titan Xp GPU, while the all-molecule dataset requires roughly 16 hours to train. Test set losses, expressed as the mean absolute error over each force component for each sample, are presented in Table 2.

Table 2: Mean absolute error of force components (in $\frac{meV}{\text{Å}}$) for geometric algebra attention networks, NequIP[31], and SchNet[9] architectures.

| Molecule | This work | NequIP | SchNet |
|---|---|---|---|
| Aspirin | $37.0 \pm 1.1$ | 15.1 | 58.5 |
| Benzene | $11.8 \pm 0.5$ | 8.1 | 13.4 |
| Ethanol | $21.4 \pm 0.5$ | 9.0 | 16.9 |
| Malonaldehyde | $30.6 \pm 1.1$ | 14.6 | 28.6 |
| Naphthalene | $23.7 \pm 1.0$ | 4.2 | 25.2 |
| Salicylic acid | $30.2 \pm 1.2$ | 10.3 | 36.9 |
| Toluene | $20.5 \pm 1.3$ | 4.4 | 24.7 |
| Uracil | $27.4 \pm 0.8$ | 7.5 | 24.3 |
| **All molecules** | $10.7 \pm 0.2$ | | |

Our geometric algebra attention networks produce results competitive with SchNet[9], an architecture using learned radial distance convolution filters. Although the models generated here do not outperform the Neural Equivariant Interatomic Potentials by Batzner *et al.*[31], we note that our models are trained for a fraction of the time (2 GPU hours and 800 epochs for our method, compared to on the order of 8 GPU days and 2500 epochs for NequIP) and without drastic hyperparameter tuning aside from optimizing the number of residual blocks to use in the network architecture. Notably, the models trained on all 8,000 molecular snapshots perform significantly better than almost all of the specialized models, indicating that additional data could likely improve the results presented here even without careful hyperparameter optimization.

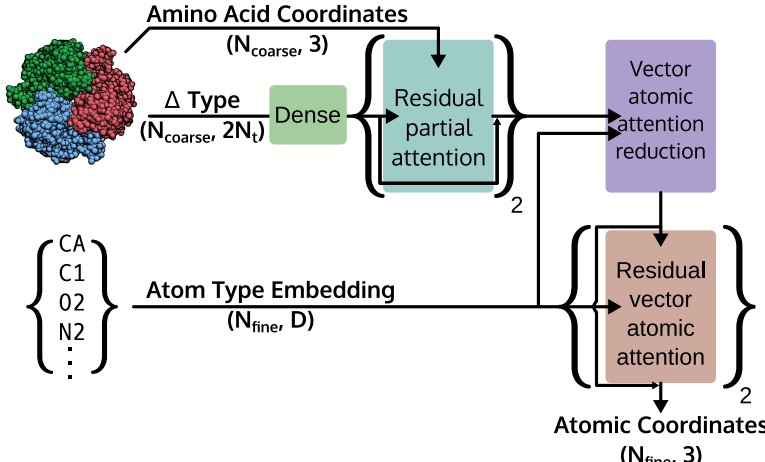

Figure 4: Network architecture for inverting a coarse-grained mapping of a protein. Models are trained to predict atomic-resolution coordinates from $N_{\text{coarse}} = 12$ neighboring amino acid centers of mass using geometrically-informed scalar-to-scalar attention (blue), scalar-to-vector attention (purple), and vector-to-vector attention (red).

## Backmapping Coarse-Graining in Proteins

When simulating large molecules—such as proteins or other polymers—it is common to employ *coarse graining*: a process by which groups of particles are merged into (fewer) distinct beads, enabling faster simulations by decreasing the number of degrees of freedom of the model[4]. Although data-driven approaches have been highly successful to formulate coarse graining operations in the forward direction (that is, from more-detailed to less-detailed systems), some problems are best solved using the original, fine-grained system coordinates, which are not directly available in coarse-grained simulations. To demonstrate the potential for our geometric algebra attention mechanism on this task, we train models to predict the coordinates of the heavy atoms that form an amino acid from the centers of mass of the nearest-neighbor amino acids. We take 19 protein structures[32–50] that have high-resolution structural refinements (with resolution error less than or equal to $1.0\,\text{Å}$) and were published between 2015 and 2020 from the Protein Data Bank[51]. For applications of this method to systems at nonzero temperature, we would expect to be better-served by using an architecture that produces distributions instead of only point values, but we disregard this here for simplicity; in other words, here we are teaching models to memorize the results of structure refinement algorithms, which may be different for each PDB entry. For every amino acid in each entry, we create a point cloud of its 12 nearest neighbor amino acid centers of mass, as well as a point cloud of the primary amino acid's atomic coordinates relative to its center of mass. Two layers of permutation-covariant geometric product attention are applied to the coarse-grained amino acid coordinates before being passed to a layer which produces a vector output according to Equation 2 by augmenting the tuple representation $v_{ijk...}$ of Equation 1 with labels corresponding to the identity of the atom that should be produced, so that the value is calculated as

$$v_{\text{atom},ijk...} = \mathcal{J}(v_{\text{atom}}, \mathcal{V}(q_{ijk...}), \mathcal{M}(v_i, v_j, v_k, ...)).$$

Following this layer—which maps coarse-grained coordinates of amino acids to fine-grained coordinates of atoms—two rotation-covariant layers are applied to the atomic coordinates to further refine them, as shown in Figure 4.

Because the resolution of the structural refinement algorithms is on the order of $0.5\,\text{Å}$ or greater, we use the training set error as a measure of the learning progress of the models instead of performing a standard split of training, validation, and test set data. After training for roughly 3 hours on

an NVIDIA Titan Xp GPU, models achieve a mean absolute error of $0.128\,\text{Å} \pm 0.002\,\text{Å}$ (down from approximately $0.5\,\text{Å}$ initially), indicating that they are able to learn to reconstruct atomic-scale coordinates from coarse-grained positions.

## Discussion

Overall, we find the architectures formulated here to be useful for a variety of tasks. Rather than being limited to operating on bond distances and angles as in SchNet[9], PhysNet[10], and DimeNet[11], geometric algebra provides a systematic way to build functions with the desired rotation- and permutation-equivariance, with the flexibility to incorporate other types of geometric objects (such as the orientation quaternion commonly used for anisotropic particles in molecular dynamics methods[52]). The attention mechanism presented here provides a simple yet powerful method to incorporate both geometric and node-level signals. The primitives of our geometric algebra attention scheme—distances, areas, angles, and volumes—and the calculated attention weights naturally lend themselves to interpretability, which we believe will prove useful in distilling insights from trained models.

### Limitations

**Combination of terms.** Although the architectures presented here work well for the problems we have selected, creating geometric products of vectors is only a subset of the valid combinations that could be generated. In these cases we have carefully chosen sums and differences of input vectors to respect symmetries we would like to impose on the system—such as using the pairwise distance of all input coordinates for the molecular force regression task to impose translation invariance—but it is possible that more powerful models could be formed by incorporating learned linear combinations of inputs or intermediate multivector quantities. We leave this as a topic of future work.

**Computational scaling and neighborhood definition.** An obvious limitation to using higher-degree correlations lies in the computational complexity and memory scaling of generating tuples, which are both proportional to $N^r$ for neighborhoods of $N$ coordinates and tuples of length $r$. Polynomial scaling behavior can be ameliorated by restricting which combinations of input points are considered, essentially treating the attention weights of all other combinations as 0. These combinations could be randomly sampled from all valid indices $ijk...$ or use more physically-relevant restrictions, such as utilizing the molecular connectivity graph for molecular force regression or edges derived from the Voronoi tessellation for other applications. If smoothness of model predictions is a concern—as may be the case for learning general N-body interaction potentials, for example—the architectures presented here could be augmented by incorporating weights that decay to 0 as bonds are broken in the Voronoi diagram graph[23].

## Conclusion

In this work, we have presented a strategy for developing rotation- and permutation-equivariant neural network architectures by combining geometric algebra and attention mechanisms. These architectures operate directly on the vector, scalar, and other geometric quantities of interest to produce outputs which respect desirable symmetries by construction. We believe that the mathematical simplicity and the insights derived from attention maps are particularly appealing aspects of the algorithms presented here. We hope that these architectures will help a wider range of scientific disciplines reap the benefits of geometric deep learning.

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
