# OpenReview forum: "Geometric Attention Networks for Small Point Clouds"
_NeurIPS.cc/2021/Conference — NeurIPS 2021 Submitted_

### Official Review · Reviewer_f3bU · 2021-07-10

**Rating:** 5
**Confidence:** 3

**Summary:**

The paper proposes a new attention mechanism based on Clifford Algebra. The main idea is computing features via geometric products, showing this would help to obtain equivariant networks to some symmetries. The method is applied to some significantly relevant tasks from physic and chemistry, with a limited quantitative validation of the method.

**Ethical Concerns:**

I do not see any particular ethical concerns in the proposed method.

**Limitations And Societal Impact:**

I do not see any particular potential negative societal impact in the proposed method.

**Main Review:**

PRO
======
1) RELEVANCE OF THE TASK \
Injecting structural invariances or equivariants into a network is an active research field and has particular relevance in a disparate set of different fields. I believe this topic would be of high interest to the NeurIPS community.

2) THEORETICAL FRAMEWORK \
Clifford Algebra has been widely used in the past, showing its generality in describing different problems. I think the theoretical foundation is sound and it could gemmate new works, adapting the framework to the need of specific communities. The paper state that the code will be available under the MIT license, which is also a contribution.

CONS
======
1) BACKGROUND AND NOVELTY \
I think the paper miss a consistent part of its positioning in the literature. Like many other works in deep learning literature, the use of Clifford Algebra was suggested a long time ago [1][2], also discussing the utility of such an approach to deal with invariants [3]. These methods have been re-proposed more recently [4][5]; similarly, other works on geometric algebra and deep learning are [6][7]. Given so, I would suggest including a section in the related work to analyze such works and emphasize the novelty of the proposed method with respect to them. At the moment, such ideas seem already well-known in the deep learning community

2) EXPERIMENTS \
The paper is a bit limited in quantitative analysis of the proposed method: \
a) In the Crystal Structure Identification is provided only the overall accuracy, without any baseline and comparison. I would suggest considering having at least some baseline that does not exploit the proposed solution (e.g., a standard MLP) to understand the benefit of using the proposed approach \
b) In the Molecule Force Regression, the results is more satisfactory; however, the networks are trained for a different amount of time (800 epochs [2 hours] for the method vs 2500 [8 days] for NequIP), and this makes it hard to understand if the comparison is fair. I would suggest training all models for a similar time, or at least add the training curves to highlight the convergence of the networks. If the proposed method converges faster, this is merit and could be emphasized. Also, why in Table 2 All molecules results and standard deviation are reported only for the method of this paper, and not for the others?\
c) In Backmapping Coarse-Graining in proteins only a mean absolute error is provided. As above, I would suggest adding some competitors and baselines.
From an analysis perspective, I think that would be good to add some studies on the behaviour of such attention, e.g., showing some examples highlighting where the proposed attention concentrate, and comparing it to a standard dot product attention mechanism. Finally, since the paper claims that different kind of symmetries can be selected by the network, I would suggest showing cases in which the attention mechanism behave in different ways and which invariant\equivariant properties could be exploited.

3) PRESENTATION \
While I think that the overall idea is clear, I think few things may be done to improve its presentation:\
a) an appendix about Geometric Algebra would be useful for the reader; such tools are not common and maybe it is useful a formal recap of the basic principles\
b) In the subsection "Attention from Geometric Products" I have difficulties following the flow of thoughts. I would suggest defining V, M, J and S by equations. This would help to understand what exactly is going on in the pipeline. Similarly, Equation 1 seems a set of passages that can be well expressed by a diagram, maybe with a graphical example. I think these would be important points that would ease a larger audience in understanding the process.\
c) I consider it important to add a better description of the architectures implementations, training losses, hyper-parameters, and everything that would help to replicate the experiments. Such information would fit well into a supplementary material document. Also, a complete analysis of time would be beneficial since (as stated in the limitations) computational complexity and memory scaling is critical. \
d) I would suggest making each task of the Results section less verbose and schematizing better its structure with some paragraphs (e.g., for each task specify: data, goal, architecture and loss details, numerical results, analysis\comment)

[1]: Back Propagation in a Clifford Algebra, Pearson and Bisset, 1992 \
[2]: Neural Networks in the Clifford Domain, Pearson and Bisset, 1994 \
[3]: A new self-organizing neural network using geometric algebra, Corrochano et al., 1996 \
[4]: On Clifford neurons and Clifford multi-layer perceptions, Buchholz and Sommer, 2008 \
[5]: An Extended Multilayer Perceptron Model Using Reduced Geometric Algebra, Li and Cao, 2019 \
[6]: Embed Me If You Can: A Geometric Perceptron, Melnyk et al., 2021 \
[7]: Knowledge Graph Embeddings in Geometric Algebras, Xu et al., 2021

MINOR QUESTIONS
===============
Concerning the technical elements of the method, I have some curiosities:\
a) by my intuition, this method is considering an attention mechanism that generalizes the standard one; this is achieved by considering a more general mathematical structure that can produce not only scalars but also other objects (vectors, bivector, trivectors). Some of these objects have quantities that have rotation invariance measures (e.g., signed area and signed volumes). Is this reasoning correct? If yes, given that in many cases, the symmetries of interest are known a priori, what is the benefit of the proposed approach w.r.t. other already invariant quantities (e.g., using the norms of point distances as features)?\
b) in the Limitations, it is reported that "geometric products of vectors is only a subset of the valid combinations that could be generated". But this is true for any proposed feature combinations; e.g., the standard MLP is just one of the many possible ways to recombine the features. So I would not mention this as a proper limitation of the method, since in general, the main approach is to stick to the simplest and more general thing instead of proposing complex features combinations. Or am I missing something?\
c) Why do you state that this method is specific for small point clouds? Is it due to the computational limitations, or do you think that such invariants are less useful in the case of larger ones?

RATING JUSTIFICATION
============
Overall I like the theory and the idea behind the method. However, it is hard to understand the correct placement of the paper among similar works, both in terms of novelty and quantitative evaluations on the proposed tasks. I am looking for the rebuttal phase.

**Time Spent Reviewing:**

6

---

> ### Author Response · Authors · 2021-08-10
> **Initial response to review**
>
> ## Review Point 1
>
> We thank the reviewer for their detailed and insightful review! Indeed, the authors had missed the previous work on Clifford algebra-structured neural networks in our literature review and it should certainly be incorporated into the discussion of past work. One important distinction between this past work and the method proposed in our current manuscript is that rotation equivariance is not guaranteed in a general multivector-valued neural network; while using multivectors as intermediates in a neural network is likely a very powerful idea for the types of applications presented here, we would need to carefully consider an appropriate attention structure that is sufficiently expressive while also enforcing the rotation equivariance which we desire.
>
> ## Review Point 2
>
> The points about baseline comparisons are well-taken; essentially, we see the structure identification task (as it is presented here, at least) as a toy problem that should be straightforward for any comparable method to solve, while the coarse grain backmapping problem was perhaps too ambitious of a task to use as a sample application and could use a more thorough treatment by having its own paper, based on applications of deep learning to this problem currently in the literature, such as [Li *et al.*](https://dx.doi.org/10.1063/5.0012320) and [An and Deshmukh](https://dx.doi.org/10.1039/d0cc02651d). As mentioned in our response to reviewer qUKR, we could certainly incorporate one or both of these methods as baselines, but the requirement to augment training data may make it difficult to evaluate exactly how well these models are able to generalize.
>
> For the molecular force regression task, we report standard errors only on our results because the other methods’ results are used as presented in their respective papers and those authors did not report errors from multiple independent runs, as far as we are aware. Although our networks seem to converge reasonably quickly, the NequIP authors use a much longer timescale with a training scheme similar to ours (the patience they use to decrease their learning rate after validation loss fails to improve is 1000 epochs, compared to our 20 epochs, for example); it is possible that simply slowing down the learning rate decrease and early stopping criterion would yield improvements.
>
> Visualizing the results of the coarse-graining back-mapping attention would be a good additional figure to include; we thank the reviewer for the idea. It is not immediately clear to us how to compare results from our geometric attention mechanism to typical dot product self-attention without creating an ad-hoc way to augment dot-product self-attention with information about geometry, but perhaps we are misunderstanding the suggestion?
>
> ## Review Point 3
>
> ### Subpoint A
>
> From feedback from this and the other reviews, it is clear that we should definitely include a section in the SI with a basic introduction to geometric algebra as it is used in this work. Thank you!
>
> ### Subpoint B
>
> We can add a diagram to illustrate the mechanism; for concreteness for the architectures presented here, for most of the example use cases presented in this manuscript $\mathcal{M}$ and $\mathcal{J}$ are learned linear projections, while $\mathcal{V}$ and $\mathcal{S}$ are MLPs with 1 hidden layer; additionally, layer normalization is applied to the output of $\mathcal{V}$.
>
> ### Subpoint C
>
> We can certainly add full details of the network architecture, training, and other hyperparameters to a supplementary information document; we hope that the python code we have included to train and evaluate the models is also helpful in this regard, but we recognize that different representations are more useful for different readers. As for a complete analysis of time, could you clarify which sort of time information you are interested in?
>
> ### Subpoint D
>
> Feedback taken; we can try to modularize the text structure more clearly.
>
> ## Minor Questions
>
> The intuition in (a) is correct; the network generates attention weights based on sets of per-bond values and rotation-invariant geometric quantities associated with those bonds. One interesting note is that we could apply a standard self-attention mechanism to each of these per-tuple representations to get attention weights, but that would increase the complexity of the algorithm to $N^{2r}$ (where $r$ is the rank of the attention—2 for pairwise, 3 for triplet-wise, and so on) instead of $N^r$. This is the reason that we introduce the separate score-generating functions $\mathcal{S}$ to generate logits for attention weights directly. Aside from being able to distinguish chiral point clouds (wherein we believe rank 3 attention would be necessary, since the signed volume reflects the chiral arrangement of points with respect to the origin), the choice of $r$ serves mostly to dictate the complexity of the interactions upon which models are built (i.e. single-value, two-value, and so on).
>
> For point (b), we agree with the argument in principle, but there are particular instances where pairwise differences of vectors may be important inputs (as well as, potentially, pairwise sums) as a special case. For example, in the molecular force regression application we first convert the representation from a world-centered coordinate system to atom-centered around each atom by calculating the pairwise vector difference of all atomic coordinates, which are then passed through the attention mechanism. This is the natural representation to gain translation invariance, which is an important physical attribute for the model. We believe that it may be useful to impart an inductive bias to make it easier for the model to find such pairwise differences or sums than to learn to approximate them from data.
>
> For point (c) the concern is mostly due to the scaling limitations; it is possible that the method could be applicable to larger point clouds, but its usefulness would likely heavily depend on exactly which application it is being used for and how neighborhoods are sampled. One slight difference is that our method is designed with the sort of high-information per-node signals commonly used in graph neural networks in mind, in contrast to many of the methods designed for large point clouds that primarily focus on geometric information.

---

> > ### Comment · Reviewer_f3bU · 2021-08-31
> > **Final Comment**
> >
> > Dear Authors,
> >
> > I would acknowledge your effort in replying; thank you for your work and answers. Indeed your clarifications have been useful to better understand your work.
> > However, I still think the experimental setting is not completely satisfactory; the aim of the paper is to show that the method is "able to solve a variety of problems", but the lack of baselines or of some golden standard (e.g., human level in the task) makes it hard to appreciate such point. Also, the extra experiment with force regression task shows that by changing the training policy the network performs dramatically better. This suggests that probably the training aspect could be a bit more investigated before publication, to report accurate results for the community and avoid "underestimation" of the method capabilities.
> >
> > Also, there is a significant problem with discussing the literature, the related positioning, and the general presentation. While I think the situation is better than the one highlighted by Reviewer yNSv, I think a significant effort should be done before publication.
> >
> > Given these points, I think the work is not ready and I will keep my initial rating. I wish you have found useful suggestions in these reviews, and I wish you good luck with your work!

---

### Official Review · Reviewer_yNSv · 2021-07-14

**Rating:** 4
**Confidence:** 2

**Summary:**

The authors propose a deep learning architecture for small point clouds that is permutation and rotation equivariant, which is based on geometric algebra. The efficiency of the method is demonstrated in the experiments.

**Limitations And Societal Impact:**

The authors discuss some limitations of the proposed idea.

As regards the potential negative societal impact, they argue that it is hard to find any since the method is agnostic to applications.

**Main Review:**

I think that the writing of the paper is rather confusing. Unfortunately, I am not able to judge the proposed idea, which might be interesting, because I could not understand it after trying to read the paper couple of times. In general, I find the clarity of the paper lacking and it is very hard to get the intuition behind the presented technicalities. In my opinion, the paper should be improved significantly, such that to be accessible from the reader.

For example:
1) This is a rather technical paper, with a lot of text. I think the mathematical context should be formulated more explicitly e.g. dimensionalities, examples of multivectors, etc.
2) The problem of interest is only presented from a high-level point of view. I think more specific examples are necessary.
3) I could not get the intuition behind the proposed modeling choice. Maybe some images and easy examples are useful to make the presented ideas accessible.
4) There is a lot of technical terminology (invariant, equivariant, covariant, multivectors, etc.), but I think is not clear exactly what each of them represents.
5) Figure 1 does not help to understand the concepts.


**Time Spent Reviewing:**

3

---

> ### Author Response · Authors · 2021-08-10
> **Initial response to review**
>
> We thank the reviewer for their feedback on these matters. We believe that adding an appendix on the fundamental components of geometric algebra used in the manuscript would likely be the best path to improve the accessibility of the method. For the purposes of discussion here, example multivectors for 3D space include $2$ (a pure scalar), $0.5 e_1 + e_2$ (a vector), $1 + 3e_1e_2 + e_1e_3$ (a scalar plus a bivector), and $-e_1e_2e_3$ (a trivector). Multivectors in 3D can have up to 8 components, corresponding to the ways to include or exclude the three basis vectors $e_1$, $e_2$ and $e3$. The product of identical orthogonal vectors is $1$ and orthogonal vectors are antisymmetric with respect to swapping: $e_1e_1 = 1$ and $e_1e_2 = -e_2e_1$. Many operations, including the geometric product, distribute over the sum used to express multivectors. We hope that this extremely brief set of ideas is a good starting point for discussion.
>
> For points 2 and 3, could we get a better understanding of which parts you feel could use more concrete examples, and which proposed modeling choice you had a hard time understanding?
>
> For point 4, in addition to the geometric algebra information to be placed in the appendix (which will explain in more detail how multivectors are defined and work), we propose to expand the text which had been relegated to the footnote of page 3 on equivariance to a section with more detail on topics of equivariance. For the purposes of clarity during review, briefly: multivectors are the objects on which geometric algebra operates (scalars and vectors are multivectors, but operations like the geometric product may generate other quantities corresponding to signed areas [bivectors] or volumes [trivectors]); a function is invariant to an operation if applying the operation to its input does not change the output, and covariant to the operation if applying the operation to its input also changes the output in the same way; and we use equivariance as a blanket term in this work to refer to the set of considerations involving invariance or covariance.
>
> We can attempt to improve Figure 1; the intent was to visually demonstrate how the rotation-invariant attributes of point clouds (that is, bond lengths, areas, and angles, while also incorporating point- or bond-level signals) map to the terms of our proposed attention mechanism.

---

> > ### Comment · Reviewer_yNSv · 2021-08-31
> > **After rebuttal**
> >
> > I would like to thank the authors for the rebuttal. Initially, I believe that the paper should be improved in its clarity in order to become more accessible. Also, I see that the other reviewers raised some additional issues. I think that in the current state the paper is not ready for publication and for this reason I will keep my score to 4.

---

### Official Review · Reviewer_qUKR · 2021-07-16

**Rating:** 5
**Confidence:** 4

**Summary:**

The paper presents a new framework for building graph neural networks on point clouds that obey invariance or covariance properties through the use of Geometric Algebra. Geometric Algebra involves the use of a binary operator called the Geometric Product that operates on geometric quantities like scalars, vectors, bivectors and trivectors.

The proposed framework involves the building neural networks that are composed of geometric products of such quantities together with self-attention. The geometric products could be designed to maintain rotation invariance or covariance as needed, and attention can impose permutation equivariance which is important for point clouds.

The paper presents experimental evidence of the effectiveness of this method on three different problems taken from physics, chemistry and biology.

**Limitations And Societal Impact:**

The authors discuss the limitations of their research and propose some future work to possibly address it.

**Main Review:**

The paper introduces the idea of Geometric Algebra within deep learning to build invariant or equivariant models for building invariant or covariant models for point clouds. This is important in many natural science problems where the point clouds are composed of atoms / molecues / physical objects that obey these properties. Past research has shown that models that obey these properties tend to be significantly more data efficient.

This paper makes use of Geometric Algebra to come up with a systematic framework for building and reasoning about such models. This is more general than methods like SchNet or Dimenet that only operate on pairwise distances and bond angles.

## Originality
The use of Geometric Algebra is truly original as per my knowledge.

## Quality
The paper presents three sets of experiments: crystal structure identification, molecular force regression and coarse graining in proteins.

For the crystal structure and protein tasks, the authors do not present any baseline methods. Without a baseline, it is difficult to understand the complexity inherent in the problem. I would recommend the authors to add at least a simple baseline.

For the force regression task , the authors compare against SchNet and Nequip. The proposed model performs comparably to SchNet, but significantly worse than Nequip (though its much faster than Nequip). It would be good to discuss why this might be the case. Is there a fundamental limitation to this method that is causing this? If not, would a simple modification / bigger model improve performance?

The authors do claim that training on all data performs better, indicating that more data could help improve their model's performance. However, this claim does not make sense without a corresponding number for Nequip, since it is possible Nequip will perform even better when trained on all data.

Without making these changes, it would be hard to definitively conclude that the presented experiments support the claims made by the paper.

Overall, this feels like a very novel work, that needs more work before it can be published.

## Clarity
The paper is clearly written and is easy to follow in spite of the authors using a new branch of mathematics.
A few minor suggestions to improve clarity further:
* Please add some pictures to explain what bivectors and trivectors mean for the uninitiated.
* In equations where you use indices `ijk...`, it may be better to use `1:n`.

## Significance
Being able to build in equivariance / invariance into NN models is extremely important for many natural science problems. This paper has the potential to impact a large number of future papers. However, the claims need to be validated more thoroughly with experiments before this can happen.


**Time Spent Reviewing:**

3

---

> ### Author Response · Authors · 2021-08-10
> **Initial response to review**
>
> We thank the reviewer for their ideas and suggestions. The suggestion for baselines on the other two tasks absolutely makes sense within the context of the manuscript. Something that we should have clarified from the start is that we expect the structure identification task to be very easy, almost at the level of a toy example; for example, a quick test of an MLP trained on rotation-invariant spherical harmonic features is able to reach around 96% accuracy on this task without even attempting any hyperparameter tuning. Our intent was to sketch out our proposed attention mechanism and a few sample applications without delving too far into the arms race of tuning hyperparameters of competing models, but we recognize that the current manuscript should include this context.
>
> In contrast to the structure identification task—which we see as one that should be trivial—the reason we did not include a baseline for the protein backmapping task is that we were not aware of any comparable machine learning-based approaches in the literature that respect the rotation and permutation symmetry of the problem: [Li *et al.*](https://dx.doi.org/10.1063/5.0012320) encode the problem as an RGB image-to-image conversion, while [An and Deshmukh](https://dx.doi.org/10.1039/d0cc02651d) simply train models directly on Cartesian coordinates. We could certainly include one or both of these methods as baselines, but we believe that it may be difficult to directly compare since these methods would likely require extensive data augmentation to learn the symmetries that are automatically encoded in the structure of our attention mechanism. Another approach is that of [Wang and Gómez-Bombarelli](https://doi.org/10.1038/s41524-019-0261-5), who learn a static linear decoder that is likely better-suited to coarse grained systems with distinguishable beads (in contrast to our protein backmapping setup, which does not label amino acids with their position within the protein sequence); because proteins often loop back on themselves, we believe this approach would be less useful in large proteins than in applications to smaller molecules.
>
> Concerning the difference in performance with respect to NequIP, one of the attributes that the authors of the SE(3) equivariant convolutions (upon which NequIP is built) emphasize is the ability of that method to propagate not only scalar values, but also vectors and other higher-order geometric values. Currently our method produces only scalars or vector values, but it may be the case that utilizing multivector values as intermediates within networks would improve performance; care would need to be taken to ensure proper rotation equivariance in the attention mechanism, but we believe it to be doable. A reasonable intermediate may be to allow the network to learn rotation-equivariant modifications to the pairwise distance vectors, which are currently used without alteration at each block of the network. Alternatively, the performance difference could simply be a matter of hyperparameter tuning: although we performed a small amount of tuning to find reasonable widths and depths of networks, we could more extensively optimize the architecture of the MLP components of the network (the $\mathcal{V}$ and $\mathcal{S}$ functions).
>
> Our intent with the “all-molecule” comparison was not to imply that NEquIP had reached its limit in performance, but rather that our model could likely perform better than the purposefully data-limited benchmarking scheme typically used for the MD17 dataset (training on 1,000 samples is under 1% of the data available in the dataset for any given molecule). We intend to modify our discussion to make this more clear.
>
> We also intend to add a supplementary information section with basics of geometric algebra, including an illustration of multivectors and their components in 3D.

---

### Official Review · Reviewer_pgXS · 2021-07-20

**Rating:** 5
**Confidence:** 4

**Summary:**

This paper studies attention from a geometric algebra perspective. Geometric algebra provides a useful tool to combine vector, scalar, and other types of geometric inputs to account for rotation invariance/equivariance and permutation invariance/equivariance. In this paper, the attention mechanism is factored into four functions: a value-generating function, a tuple value-merging function, a joining function that summarizes the invariant and tuple representations, and a score-generating function. This new perspective on attention is novel and can motivate more follow-up works on studying high-order attention. On the other hand, however, this paper doesn't introduce new module design beyond interpreting attention and the actual implementation is the same with self-attention.

**Limitations And Societal Impact:**

1. My major concern is the difference between the actual method and the self-attention. Although this paper proposes a good interpretation and motivation to study high-order attention, it does not propose a new module. The implementation is actually an attention module. If this is the major contribution of this paper, what is the new component?
2. Following 1, I would suggest authors to discussion the difference from Set Transformer and Point Transformer. Essentially, this paper uses exactly the same operation as Set Transformer.
3. In the experiments, this method only compares to NequIP and SchNet. Why not try PointNet/DGCNN? I understand this paper claims that PointNet/DGCNN work on "large" point clouds but they are actually applicable to the "small" point cloud case.
4. Even compared to NequIP and SchNet, this paper doesn't show consistent improvement, which makes me even worry about the empirical value of the proposed method.

**Main Review:**

1. This paper proposes a novel view on point-based attention. This new perspective is based on geometric algebra and is easy to generalize to high-order case. I think this is a great attempt to incorporate many useful mathematical tools into a machine learning model.
2. The rotation equivariant/invariant features are easy to obtain under this new construction. These features are directly related to the corresponding components of multivectors.
3. Application-wise, this paper tackles important molecule identification/regression problems.


======================Post rebuttal====================================

I thank authors for providing response to my questions. However, my concerns are not fully addressed: 1. the performance is worse than the baselines; 2. the learned score function is essentially similar to the the one (dot-product) used in multi-headed attention except this paper uses a "relation-net" style formulation. So I am leaning to rejection of this paper.

**Time Spent Reviewing:**

2

---

> ### Author Response · Authors · 2021-08-10
> **Initial response to review**
>
> We thank the reviewer for their valuable feedback. In particular, to address limitations 1 and 2 mentioned in the review, we intend to more clearly outline the difference between our proposed attention mechanism and attention mechanisms commonly used in the literature, as well as between set and point transformers. Although it would be possible to come close to standard attention mechanisms using a geometry-derived embedding with our pairwise attention and particular choices of functions $\mathcal{S}$, $\mathcal{M}$, and $\mathcal{J}$, our formulation is somewhat more general and, in typical usage, the methods will not be equivalent. In particular, it is unclear to the authors that it is straightforward to create expressive functions that are also rotation equivariant solely through an embedding with standard dot-product self-attention. The key features of our attention mechanism, as we see them, are (1) incorporating geometry of input points as a first-class element in the attention mechanism; (2) the ability to impart desired rotation and permutation equivariance simultaneously; and (3) being able to dictate the order of correlations on which the model operates, for example by changing between using pairwise attention or triplet-wise attention. While it would be possible to use standard pairwise attention on the rotation-invariant representations of each pair, we point out that this would lead to much worse scaling in the size of the neighborhood $N$: $N^4$ to capture pairwise correlations, $N^6$ for triplet-wise attention, and so on. Instead, we utilize a learnable score function $\mathcal{S}$ that generates attention logits directly from the representation of each tuple. While the set and point transformers can both work on point clouds, as far as the authors are aware they make no effort to deal with rotation equivariance, which is crucial for correctness in the molecular force regression task especially. We should certainly include the set and point transformers in our discussion of previous work.
>
> For point 3, to our understanding PointNet and DGCNN only operate on the coordinates of points—without incorporating any GNN-like signal vertices or edges—as is crucial for all three applications presented in this manuscript, so we would expect the performance of those architectures to be poor without modifications. The simplest example to demonstrate this is in the crystal structure identification task, where $cI2$-W and $cP2$-CsCl are identical in geometry but have different “colorings” of points, as do $cF8$-C and $cF8$-ZnS.
>
> In terms of point 4, our intention when presenting these architectures was more to demonstrate that the geometric algebra-based attention mechanism worked to incorporate both node-level and geometric-level signals to be able to solve a variety of problems, rather than focus on any particular benchmark dataset; to that point, to our knowledge, standardized benchmarks do not even exist for two out of the three sample tasks (structure identification and coarse-grain backmapping) presented here. We believe that careful hyperparameter optimization and longer training times would significantly improve the results presented for the molecular force regression task, but even without as much optimization we believe that results competitive with SchNet demonstrate that these methods are worth pursuing.

---

### Author Response · Authors · 2021-08-15
**Additional Experiments for Force Regression Task with Slower Training Schedule**

Following the discussion from reviewers around comparisons of our method with others on the molecular force regression task, we have trained new models with the same architecture but using the significantly slower training schedule used by the authors of the NequIP paper: decreasing the learning rate by 0.8 after 1000 epochs of no improvement in the validation loss and ending training after 2500 epochs of no improvement, or 50000 epochs in total (previously our values during training were a factor of 0.75 and epoch durations of 20, 50, and 800, respectively). Below are the results from Table 2 including data from individual longer-duration runs (we intend to run multiple replicas to obtain error estimates for these values as well).

|    Molecule    	| This work (fast schedule) 	| This work (slow schedule) 	| NequIP 	| SchNet 	|
|:--------------:	|:-------------------------:	|:-------------------------:	|:-------:	|:-------:	|
|     Aspirin    	|            37.0           	|  12.6                         	|   15.1  	|   58.5  	|
|     Benzene    	|            11.8           	|  6.5                         	|   8.1   	|   13.4  	|
|     Ethanol    	|            21.4           	|       4.8                    	|   9.0   	|   16.9  	|
|  Malonaldehyde 	|            30.6           	|    8.7                       	|   14.6  	|   28.6  	|
|   Naphthalene  	|            23.7           	|   4.2                        	|   4.2   	|   25.2  	|
| Salicylic acid 	|            30.2           	|        7.8                   	|   10.3  	|   36.9  	|
|     Toluene    	|            20.5           	|                 4.5          	|   4.4   	|   24.7  	|
|     Uracil     	|            27.4           	|                    6.2       	|   7.5   	|   24.3  	|
|  All molecules 	|            10.7           	|                           	|         	|         	|

In the end, it seems there were significant performance improvements that could be extracted simply by training longer; intuitively, we believe that it makes sense for our attention-based method to outperform the convolutions used in the NequIP work. We are happy to answer any follow-up questions that these additional experiments prompt by the reviewing committee and, once again, thank them for their time and careful consideration.

---

### Decision · Program_Chairs · 2021-09-27

**Decision:**

Reject

**Comment:**

Thank you for the rebuttal. Unfortunately, the paper is rejected from NeurIPS 2021.
The reviewers' main criticism was the lacking exposition of the method in the paper, and relation to previous work. I would recommend the authors to improve their exposition of "Geometric Algebra" and the clarity of their respective architectures. Reviewers also complained that some baselines are required to better evaluate the performance of the method in practice.